# Association between the Dietary Phytochemical Index and Lower Prevalence of Obesity in Korean Preschoolers

**DOI:** 10.3390/nu15112439

**Published:** 2023-05-24

**Authors:** Ye-Ji Han, Jung-Hyun Baek, Seong-Kwan Jung, Joshua SungWoo Yang, Na-Rae Shin, Mi-Young Park

**Affiliations:** 1Department of Food and Nutrition, Sungshin Women’s University, Seoul 01133, Republic of Korea; yeji.h1n@gmail.com; 2Department of Pediatrics, Woori Children’s Hospital, Seoul 08291, Republic of Korea; backjh@woorisoa.co.kr (J.-H.B.); jungsk@woorisoa.co.kr (S.-K.J.); 3Healthcare Development Head, R&D Center, NGeneBio Inc., Seoul 08390, Republic of Korea; sungwoo.yang@ngenebio.com (J.S.Y.); narae.shin@ngenebio.com (N.-R.S.); 4Institute of Health and Environment, Graduate School of Public Health, Seoul National University, Seoul 08826, Republic of Korea

**Keywords:** dietary phytochemical index, obesity, preschooler, child, Korea National Health and Nutrition Examination Survey (KNHANES)

## Abstract

Little is known regarding Korean preschooler dietary phytochemical index (DPIs). We used the 24 h recall data of 1196 participants aged 3–5 years from the Korea National Health and Nutrition Examination Survey to study the association between dietary food intake and obesity prevalence. The amount of dietary intake by food group was compared according to sex and DPI quartile. Multivariable-adjusted odds ratios (ORs) and 95% confidence intervals (CIs) were calculated using logistic regression models. The average total DPI and energy from phytochemical food groups were not significantly different according to sex, although boys had a higher total daily food intake. Different inclinations between DPI quartiles and amount of intake were observed in the food groups; specifically, beans showed a higher intake difference between Q1 and Q4 for boys than in the other food groups. The highest DPI quartile had a significantly lower obesity prevalence than the lowest DPI quartile in all models for boys only when obesity prevalence by weight percentile was analyzed (Model 3, OR: 0.287, 95% CI: 0.095–0.868, *p* for trend < 0.05). Our results suggest a high DPI could help prevent obesity in preschoolers.

## 1. Introduction

An increasing trend in childhood obesity and overweight has been observed worldwide [1,2,3]. The prevalence of obesity among children aged 2–18 years has increased from 8.6% in 2001 to 9.8% in 2017 in Korea [4]. Many studies have demonstrated childhood obesity to be related to an increased risk of several adverse health outcomes at an early age, including cardiovascular disease, fatty liver infiltration, and sleep apnea [5,6,7]. Moreover, controlling childhood obesity is very important since childhood obesity has a high risk of developing into adult obesity [8,9]. A follow-up study found that overweight children aged 2–5 years had >4 times greater tendency to become overweight adults [10]. Childhood obesity reportedly increases the risk of cardiovascular disease, cancer [11], and mortality [5,12,13] in adulthood.

The increase in childhood obesity in Korea seems to be the result of a decrease in vegetable intake and an increase in animal food product intake due to rapid modernization and a westernized diet. The traditional food culture of Koreans includes vegetable-based proteins, such as tofu made of soybeans [14]. However, the proportion of individuals consuming more than 500 g vegetables and fruits daily decreased by approximately 10% from 36.5% in 2011 to 26.2% in 2020 in Korea [15]. 

Previous studies have demonstrated that sufficient intake of vegetables, fruits, and whole grains, which are representative phytochemical food groups, helps in weight control [16,17,18]. Phytochemicals, which are active substances present mainly in plant foods, protect the body from oxidative stress and free radicals with antioxidant functions [19].

The dietary phytochemical index (DPI) is defined as the percentage of dietary calories derived from foods rich in phytochemicals [20]; a higher DPI is strongly associated with a lower prevalence of overweight/obesity [21,22,23], cardiovascular disease, metabolic syndrome [24], and cancer [20]. A meta-analysis revealed that a lower DPI decreases the risk of overweight/obesity by 19% (95% confidence interval [8]: 0.74–0.90) [21]. However, the results demonstrated significant heterogeneity according to age, sex, and obesity criteria [21].

Therefore, studying the amount of intake by food group and DPI among Koreans is necessary. Nevertheless, few studies exist on the DPI of Korean preschoolers. Moreover, studies that focused on children did not consider sex differences. This study investigated the association between the DPI and prevalence of obesity/overweight among Korean preschoolers according to sex using the 2013–2018 data from the Korea National Health and Nutrition Examination Survey (KNHANES).

## 2. Materials and Methods

### 2.1. Data Source and Study Population

We used survey data from over 6 years (2013–2018) from the KNHANES. The KNHANES is a large-scale cross-sectional survey conducted among individuals aged ≥1 year by the Korea Centers for Disease Control and Prevention (KDCA) since 1998. The nationwide cross-sectional study was conducted to identify health-related factors with a health examination, health interview, and nutritional survey among approximately 2000–3000 South Koreans per generation. The KNHANES was approved by the Institutional Review Board of the KDCA (Approval numbers: 2013-07CON-03-4C, 2013-12EXP-03-5C, and 2018-01-03-P-A). For children aged 3–5 years, written informed consent was obtained from one of the parents to use and analyze their data.

Among children aged 3–5 years (n = 1751) who participated in the survey during 2013–2018, we excluded those using the following criteria from the analyses: (1) did not participate in the 24 h recall survey (n = 61), (2) had an extreme value in the total daily energy intake (≥3000 kcal) (n = 11), (3) missing values (weight, weight percentile, height, body mass index (BMI), BMI percentile, sex, education level of the mother, education level of the father, household income, and residential area) (n = 88), and (4) had asthma, diabetes, congenital heart disease, urinary tract infection, and pneumonia (n = 395). Finally, a total of 1196 participants (boys = 623, girls= 573) were included in the study. This study was approved by the Institutional Review Board of the Seoul National University (E2108/002-002) for analyzing the secondary processing data of KDCA.

### 2.2. Demographic Data, Anthropometric Measurement, and Diagnosis of Obesity

General characteristics, such as age, weight, height, BMI, education level, household income, and residential area of the participants, were described using the basic variables, health survey, and screening survey data of the KNHANES. Considering that the participants of our study included 3–5-year-old preschoolers, we presented both the age in years and number of months to denote the age of the participant. The education levels of the mothers and fathers were considered in place of the participants’ education level and classified into three categories (middle school or lower, high school, and university graduation or higher). Household income level was classified in quartiles (low, middle-low, middle-high, and high) by the KNHANES with different standards for quartiles provided according to the survey conducted years [25]. The residential areas were classified into two categories (urban and rural).

Health surveys, including screening surveys, were conducted by trained staff according to a standardized protocol [26]. The weight and height were measured using a standardized height and scale, with the participants taking their shoes and socks off, wearing a test gown, and standing upright on a scale. Weight percentile (%) and BMI percentile (%) were provided as categorical variables by the KNHANES, using the percentiles by the growth chart for children and adolescents in 2017 as reference [26]. The growth chart is an important indicator for evaluating the growth status of children and adolescents who are in the process of growth, unlike adults. To evaluate the growth status of children and adolescents, such as short stature, underweight, and obesity, the Korea Centers for Disease Control and Prevention and the Korean Academy of Pediatrics have established and published a growth chart every 10 years since 1967 [27].

The weight percentile (%) was divided into three groups (<5%, 5–95%, >95%), and the BMI percentile (%) was divided into four groups (<5%, 5–85%, 85–95%, >95%). We defined obesity as a percentile >95% for both the weight and BMI percentiles according to the Korean Society for the Study of Obesity criteria [27].

### 2.3. Nutritional Survey Data and DPI

To calculate the DPI for children aged 3–5 years in Korea, we used the data of daily nutrient intake based on the 24 h recall survey from the KNHANES. The 24 h recall survey was conducted through face-to-face interviews with trained staff. In the case of children younger than 8 years of age, one of the parents answered the dietary recall [25]. Supplementary materials were also used to enhance recall skills and collect specific data on the survey items during the survey.

The DPI was used to calculate the approximate value of phytochemical intake from diets using the intake level of plant foods [20]. It was also useful for evaluating the health effects of the overall plant foods consumed; the calculation formula was as follows:$$DPI=\frac{Daily\;energy\;intake\;derived\;from\;phytochemical\;rich\;foods\;(\mathrm{kcal})}{Tatal\;daily\;energy\;intake\;(\mathrm{kcal})}\times 100$$

In this study, a modified version of the DPI for the Korean diet was used to calculate the DPI in Korean children aged 3–5 years [23,24]. The food groups included in the DPI were whole grains, beans, seeds and nuts, vegetables, mushrooms, and fruits. All the food groups were classified according to KNHANES. Beans included soybeans and products made from beans such as tofu. Grains and their products were divided into two groups based on whether they contained phytochemicals. Whole grains, such as oats, millet, buckwheat, whole wheat, barley, sorghum, corn, and brown rice, are rich in phytochemicals, whereas refined grains are not.

### 2.4. Statistical Analyses

Stratification variables, colony variables, and weight were all considered during the statistical analyses, as this study used data from the KNHANES, constructed through the complex sample design method. All the analyses were performed using the PROC SURVEY procedure. General characteristics of the participants according to sex were presented as mean and standard error or frequency and percentage (%).

In the analysis of intake amount by food group according to the DPI, the range of total DPI was presented as minimum–maximum, with the mean DPI of each quartile according to sex. Energy from the phytochemical food groups (kcal/day) and dietary intake amounts by the food group were presented as mean and standard error, according to the sex and DPI quartile. Differences between the sex in the total DPI, energy from the phytochemical food groups, and amount of intake by food groups were also shown by using the PROC SURVEYREG procedure to calculate *p* values.

Weight (kg), BMI (kg/m^2^), and daily energy intake (kcal/day) were presented as the mean and standard error by the DPI quartile for each sex. The *p* values were calculated using the PROC SURVEYREG procedure. Multivariable logistic regression analyses were used to calculate the odds ratios (ORs) and 95% CIs to analyze the association between DPI and obesity. Three covariate models, except for the crude model, were evaluated by adjusting for the confounding factors as follows: Model 1 was adjusted for energy intake, Model 2 was adjusted for age, and Model 3 was adjusted for energy intake and age. The *p* for trend in the DPI quartiles using linear regression analysis was evaluated using the median value of the category as a continuous variable. All data analyses were performed using the Statistical Analysis System version 9.4 software (SAS Institute, Cary, NC, USA). A *p*-value < 0.05 was considered statistically significant.

## 3. Results

### 3.1. General Characteristics

The general characteristics of the participants stratified by sex are presented in Table 1. The average age of the participants was 4.02 years (53.75 months), and no statistical difference was observed between the sexes. Boys had a higher average height and weight than girls (*p* < 0.01); significant difference between the height according to sex was shown among participants aged 4 and 5 years old (*p* < 0.01). Weight was different only in the participants aged 3 years old when stratified by age and sex. The average daily energy intake of the participants was 1370.65 kcal/day, and boys had a significantly higher energy intake than girls (*p* < 0.001). No statistical difference in the education levels of the mother and father, household income, and residential area was observed according to sex.

### 3.2. Association between the DPI and Dietary Intake

Table 2 compares the energy intake of the food group across the quartiles for sex and DPI. The mean ± standard error of total DPI according to sex was 15.43 ± 1.07 and 16.49 ± 0.45 for boys and girls, respectively. Boys had a significantly higher amount of total daily food intake (g/day) than girls; however, the total DPI and energy from the phytochemical food groups (kcal/day) were not significantly different from those of girls. The intake of refined grain, beans, milk and dairy, meat, and sugars was significantly higher in boys than in girls. Upon dietary intake analysis according to the DPI by sex, a higher DPI was associated with the amount of daily food intake in most of the food groups. Beans demonstrated a higher difference in intake between Q1 and Q4 in boys than the other food groups. Eggs, fish, and shellfish showed no significant differences in the amount of daily intake between sexes. Daily intake of meat in boys increased according to the DPI quartile (*p* < 0.01); however, no significant difference was observed in girls. No statistical difference was observed in the intake of milk, dairy, and sugars across the DPI quartiles in boys; in contrast, they significantly increased across the DPI quartiles in girls.

### 3.3. Association between DPI and Obesity

Table 3 shows the ORs for obesity according to the DPI by sex. Obesity was defined using weight and BMI percentiles. Different results were observed according to sex. Although no significant differences were observed in the weight, BMI, and daily energy intake across the DPI quartile in each sex, an association between the DPI and obesity was observed in boys when analyzing the prevalence of obesity by the weight percentile. Boys in the highest DPI quartile showed a significantly lower prevalence of obesity than those in the lowest DPI quartile in all the models (Model 3, OR: 0.287, 95% CI: 0.095–0.868, *p* for trend <0.05). In contrast, no statistically significant association was observed between the DPI and obesity prevalence, according to the BMI percentile in boys. No significant association was observed between the DPI and obesity prevalence in girls.

## 4. Discussion

This study was conducted to determine the relationship between DPI and obesity in children aged 3–5 years in Korea based on their sex. The prevalence of obesity was more than 70% lower in the group Q4 with low DPI in boys than in the group Q1 (Table 3).

Childhood obesity has a high risk of developing into adult obesity; therefore, controlling childhood obesity is very important [8,9]. According to a study that proactively tracked children’s BMI status until adolescence, 83% of 4-year-old children with obesity were overweight or obese in adolescence, and only 17% returned to normal weight [28]. Dietary factors were identified as significant predictors of the weight of preschool children [28], characterized by high consumption of energy-dense snack foods and meals in overweight and children with obesity and insufficient fruit and vegetable intake [29,30]. Food preferences and eating habits formed during preschool do not change and can result in obesity in adults [31,32]; thus, this study provides important basic data emphasizing the importance of phytochemical-rich meals for children during this period.

Many studies have highlighted the health benefits of consuming foods rich in phytochemicals. In particular, studies on obesity have focused on the metabolic function of specific phytochemical components; for example, a study exists on obesity associated with reducing fat and body fat through mechanisms such as regulating fat metabolism of polyphenols [33,34], inhibiting fat cell proliferation, increasing fatty acid oxidation of resveratrol [35], and reducing fat accumulation in the body of flavonoids [19]. However, quantifying phytochemicals in food sources is impractical in terms of cost and time because individuals consume food composed of food ingredients, not individual nutrients.

The phytochemical index (PI), proposed by McCarty, defines [20] the energy obtained from plant foods as a percentage of the total daily energy intake divided by the total daily energy intake. Foods rich in phytochemicals include whole grains, fruits, vegetables, nuts, and beans; studies on the association between PI and diseases related to the intake of such foods have been consistently published [36,37,38]. A meta-analysis summarized the relationship between DPI and the risk of overweight/obesity and demonstrated that a high phytochemical index was associated with a reduced risk of overweight/obesity [21]. In a cross-sectional study by Vincent et al. [39], significant adverse correlations were found between the DPI scores, BMI, waist circumference, waist circumference–hip ratio, and body fat rate in 54 American adults aged 18–30 years. Even in a cross-sectional study in Iran [40], the risk of abdominal obesity was reduced, regardless of the age, sex, and total energy intake at higher DPIs in relation to the DPI and metabolic syndrome. A study using KNHANES 2008–2018 data on Korean women [22] demonstrated that the prevalence of obesity and abdominal obesity decreased in women who consumed increased amounts of phytochemical-rich foods. 

The correlation between DPI and obesity has mainly been studied in adults, and few papers targeting children, especially preschoolers, exist. A cross-sectional study of 356 school-aged children (7–10 years old) in Iran demonstrated a negative relationship between phytochemical intake and obesity [23]. Overweight/obesity odds were 0.47 (0.25–0.87, *p* for trend = 0.02) for DPI Q1(14.25 ± 4.13) and Q4 (61.52 ± 16.47). A study of children and adolescents aged 6–18 years similarly revealed significantly decreased trends in BMI, hip circumference, and neck circumference for overweight and obese students based on DPI quartiles [41]. Both studies were consistent with our findings that higher phytochemical intake reduced the risk of obesity.

Our study was the first to analyze the relationship between DPI and obesity in preschoolers stratified by sex. The association was analyzed separately by sex, and significant differences were observed in the daily food intake and calories between boys and girls. Boys consumed more calories per day (kcal/day) than girls did (*p* < 0.085). Sex had no substantial impact on either weight or BMI. Our findings demonstrated that DPI quartiles and obesity prevalence only inversely correlated in boys (*p* for trend <0.05). Boys had a higher intake of beans than girls. Moreover, the consumption of meat, milk, and dairy products, the non-phytochemicals food group, was high. Despite being significant sources of protein, they did not exhibit a linear pattern, according to DPI. Thus, we only focused on the anti-obesity effect of beans as a phytochemical-rich food.

Many studies have demonstrated that obesity and metabolic syndrome increase as soy intake decreases in women [42,43]. As isoflavones have a positive effect on women’s abdominal obesity reduction due to structure similarity to that of women’s sex hormones [44,45], many studies have recommended increased protein intake through beans. As children aged 3–5 years old, who were included in this study, do not yet possess physiological differences according to sex hormones, the anti-obesity effect in boys was presumably associated with high bean intake and clear differences according to the DPIs.

The prevalence of obesity was significantly lower in boys and the high DPI group. There was no significant difference in the participant’s weight, BMI, and total energy intake; the average intake of milk, dairy, and meat groups was higher than that of girls; and soybean intake was higher among the food groups belonging to DPI. Therefore, to prevent obesity from spreading in children and adulthood, it is important to focus on the composition of the food group consumed rather than the calories. Gunther et al. [45] found that increased protein intake from dairy products was associated with BMI in the first 12 months and increased body fat at 7 years of age. Other studies have also considered dairy consumption in growing children and adults to protect against overweight and obesity [46,47]. A study of children 2–3 years old in Australia [48] found that there were associations between microbiota composition and dairy- and plant-based foods such as fruits, vegetables, beans, pulses, and nuts, indicating a link to microorganisms when analyzed based on food intake rather than nutrient intake. Therefore, in children’s nutritional research, it is recommended to consider the intake of food groups first because melas are made up of food, not nutrients. The results of this study, considering the relationship between the phytochemical intake and obesity, indicated the DPI is considered a useful tool, and investigating the composition of the DPI and microbiome in the future could be a good strategy.

This study had several limitations. First, as this was a cross-sectional study using KNHANES data, in addition to the effect of DPIs on obesity development, the inverse correlation between obesity and food intake cannot be excluded. Other environmental factors such as genetics, breastfeeding, and lifestyle, which may have affected the relationship between DPI and obesity prevalence, were also not considered. However, the participants of this study were in the process of controlling microbial communities, unlike infants and toddlers, in whom breastfeeding has a significant health impact [49]. Second, the 24 h recall used to collect dietary data had a recall bias. However, in this study, as the average value of the group was evaluated, errors that could have been caused by recall bias were balanced. Third, this study comprised a sample of preschool-aged children within a limited age range in Korea. The generalizability of the findings of other regions of children with different dietary habits remain to be elucidated. Nevertheless, this is the first study to analyze the DPI of Korean children according to sex differences. Few studies have been conducted on DPI in preschoolers. We did not limit the diagnostic criteria for obesity to BMI percentiles, as we also used weight percentiles. Therefore, our study could serve as a steppingstone for establishing the relationship between children’s DPIs and obesity in the future, which will also aid in nutrition education and formulating policies for parents.

## 5. Conclusions

In conclusion, in Korean boys aged 3–5 years, a direct correlation between high DPI and low obesity prevalence was discovered. Research should be conducted concerning the nutrition food group intake to corroborate these findings and clarify why the sex of the preschoolers’ results varied in outcomes. Future studies are warranted to investigate the prevalence of obesity to modifications in microbiome composition attributed to the consumption of phytochemical-rich foods.

## Figures and Tables

**Table 1 nutrients-15-02439-t001:** General characteristics stratified by sex among the participants.

	Total (n = 1196)	Boys (n = 623)	Girls (n = 573)	*p* Value *
Age (months)	53.75 ± 0.32	54.05 ± 1.16	53.42 ± 0.47	0.366
Age (years)	4.02 ± 0.02	4.04 ± 0.09	3.99 ± 0.04	0.360
3	386 (32.30)	203 (32.60)	183 (31.90)	0.088
4	392 (32.80)	193 (31.00)	199 (34.70)	
5	418 (34.90)	227 (36.40)	191 (33.31)	
Energy Intake (kcal/day)	1370.65 ± 14.97	1453.30 ± 48.78	1276.06 ± 20.30	<0.001
Height (cm)	105.75 ± 0.23	106.4 ± 0.82	105.0 ± 0.33	0.004
Aged 3	98.63 ± 0.60	98.86 ± 0.84	98.36 ± 0.35	0.307
Aged 4	105.86 ± 0.58	106.48 ± 0.75	105.27 ± 0.29	0.008
Aged 5	112.41 ± 0.24	113.04 ± 0.85	111.55 ± 0.37	0.002
Weight (kg)	17.59± 0.11	17.88 ± 0.37	17.26 ± 0.15	0.003
Aged 3	15.22 ± 0.36	15.47 ± 0.40	14.93 ± 0.16	0.024
Aged 4	17.52 ± 0.36	17.74 ± 0.46	17.31 ± 0.18	0.125
Aged 5	19.91 ± 0.16	20.15 ± 0.57	19.59 ± 0.24	0.088
BMI percentile (%)				
<5%	114 (9.53)	61 (9.79)	53 (9.25)	0.843
5–85%	934 (78.09)	486 (78.01)	448 (78.18)	
85–95%	77 (6.44)	37 (5.94)	40 (6.98)	
>95%	71 (5.94)	39 (6.26)	32 (5.58)	
Weight percentile (%)				
<5%	90 (7.53)	53 (8.51)	37 (6.46)	0.404
5–95%	1037 (86.71)	534 (85.71)	503 (87.78)	
>95%	69 (5.77)	36 (5.78)	33 (5.76)	
Education Level of Mother				
≤Middle school	35 (4.34)	18 (4.88)	17 (3.76)	0.717
≤High school	280 (28.66)	139 (28.03)	141 (29.34)	
≤College	700 (67.00)	361 (67.10))	339 (66.90)	
Education Level of Father				
≤Middle school	18 (2.39)	10 (2.13)	8 (2.69)	0.474
≤High school	208 (26.73)	102 (24.97)	106 (28.73)	
≤College	567 (70.88)	302 (72.90)	265 (68.58)	
Household income				
Low	82 (7.16)	41 (6.79)	41 (7.58)	0.883
Middle-low	377 (32.05)	194 (31.44)	183 (32.76)	
Middle-high	421 (34.81)	218 (35.76)	203 (33.73)	
High	316 (25.97)	170 (26.01)	146 (25.93)	
Residential area				
Urban	1018 (86.83)	533 (87.50)	485 (86.06)	0.422
Rural	178 (13.17)	90 (12.50)	88 (13.94)	

BMI, body mass index. All the values are presented as mean ± standard deviation or n (%). * *p* values were calculated using the PROC SURVEYREG.

**Table 2 nutrients-15-02439-t002:** Comparison of the dietary intake according to the dietary phytochemical index quartile by sex.

	Total (n = 1196)			Boys (n = 623)					Girls (n = 573)				
	Boys (n = 623)	Girls (n = 573)	*p* * Value	Q1 (n = 155)	Q2 (n = 156)	Q3 (n = 156)	Q4 (n = 156)	*p* Value	Q1 (n = 143)	Q2 (n = 143)	Q3 (n = 144)	Q4 (n = 143)	*p* Value
Total DPI	15.43 ± 1.07	16.49 ± 0.45	0.085	5.12 ± 0.25 (0–8.71) ^#^	11.30 ± 0.13 (0–8.71)	17.09 ± 0.16 (13.94–20.53)	28.33 ± 0.69 (20.5–59.44)	<0.001	5.71 ± 0.22 (0.01–9.22)	11.88 ± 0.13 (9.32–14.76)	17.93 ± 0.17 (14.79–21.72)	29.64 ± 0.70 (21.74–51.45)	<0.001
Energy from Phytochemical Food groups (kcal/day) ^†^	225.6 ± 15.44	208.0 ± 6.08	0.060	68.84 ± 3.92	159.04 ± 4.67	255.36 ± 8.08	386.80 ± 15.06	<0.001	73.21 ± 3.61	150.05 ± 5.06	212.55 ± 6.56	350.28 ± 12.09	<0.001
Amount of Intake by Food groups (g/day)													
Total	1059 ± 40.30	940.2 ± 16.53	0.005	874.5 ± 96.67	1008 ± 95.75	1158 ± 97.84	1197 ± 43.73	<0.001	826.1 ± 98.47	929.2 ± 97.40	940.6 ± 97.13	1056 ± 43.22	0.001
Phytochemical food groups													
Whole grains	18.43 ± 1.10	17.60 ± 1.07	0.579	7.36 ± 1.28	12.18 ± 1.02	22.53 ± 2.27	31.91 ± 2.98	<0.001	4.88 ± 0.48	13.03 ± 1.51	21.20 ± 1.91	30.55 ± 2.77	<0.001
Fruits	196.8 ± 10.69	180.6 ± 9.29	0.243	36.51 ± 4.96	132.3 ± 10.44	231.2 ± 15.72	389.1 ± 28.79	<0.001	55.11 ± 6.15	127.1 ± 10.42	183.7 ± 14.72	346.0 ± 27.33	<0.001
Beans	26.78 ± 2.92	18.56 ± 2.58	0.036	5.17 ± 0.79	17.67 ± 3.14	30.77 ± 6.99	53.77 ± 8.32	<0.001	5.29 ± 1.31	12.18 ± 2.03	15.34 ± 1.94	40.02 ± 8.99	<0.001
Seeds and Nuts	1.73 ± 0.20	2.26 ± 0.39	0.212	0.62 ± 0.11	1.74 ± 0.28	1.98 ± 0.53	2.59 ± 0.48	<0.001	0.40 ± 0.06	1.41 ± 0.31	2.06 ± 0.47	5.02 ± 1.36	<0.001
Vegetables	95.88 ± 3.38	96.08 ± 3.68	0.966	69.49 ± 5.90	92.21 ± 6.21	112.36 ± 7.33	109.62 ± 7.87	<0.001	71.39 ± 5.33	99.73 ± 7.87	106.99 ± 7.63	105.51 ± 8.06	0.001
Mushrooms	4.06 ± 0.56	3.73 ± 0.49	0.660	2.15 ± 0.42	3.34 ± 0.56	4.16 ± 0.83	6.60 ± 1.84	0.022	2.08 ± 0.59	4.01 ± 1.30	5.32 ± 0.99	3.52 ± 0.73	0.015
Non-phytochemical food groups													
Refined grains	191.2 ± 3.99	172.0 ± 4.00	0.001	196.35 ± 7.60	214.31 ± 8.16	192.11 ± 7.64	160.68 ± 7.38	<0.001	190.14 ± 9.49	186.00 ± 8.07	163.43 ± 5.92	149.59 ± 7.35	<0.001
Milk and Dairy	244.3 ± 10.19	218.2 ± 8.23	0.039	263.9 ± 22.17	258.2 ± 18.86	250.0 ± 17.43	204.6 ± 16.16	0.059	234.5 ± 18.94	254.7 ± 13.80	214.1 ± 18.09	171.4 ± 12.72	<0.001
Meats	63.91 ± 3.62	48.71 ± 2.54	<0.001	67.71 ± 6.73	58.06 ± 4.90	83.02 ± 11.08	47.64 ± 4.19	0.002	50.98 ± 4.34	54.15 ± 5.52	45.27 ± 4.91	44.49 ± 3.68	0.395
Eggs	30.51 ± 1.87	26.02 ± 1.61	0.061	33.39 ± 3.57	32.72 ± 4.57	30.11 ± 3.51	25.76 ± 3.22	0.392	32.50 ± 4.10	26.67 ± 2.76	20.95 ± 2.29	24.03 ± 3.10	0.084
Fish and Shellfish	42.43 ± 2.91	39.10 ± 2.97	0.393	33.19 ± 4.33	39.11 ± 4.70	49.61 ± 5.92	48.03 ± 6.52	0.067	39.43 ± 7.14	41.58 ± 4.99	38.80 ± 4.74	36.69 ± 6.71	0.945
Sugars	11.11 ± 0.96	7.53 ± 0.72	0.003	13.92 ± 2.49	12.16 ± 2.15	10.20 ± 1.68	8.10 ± 1.50	0.162	11.38 ± 2.03	6.91 ± 1.22	7.45 ± 1.63	4.64 ± 0.65	0.005

DPI, dietary phytochemical index; Q, quartile. All the values are presented as mean ± standard deviation. * *p* values were calculated using PROC SURVEYREG. ^#^ The range of DPI is presented as (minimum–maximum). ^†^ Includes whole grains, beans, seeds, nuts, vegetables, mushrooms, and fruits.

**Table 3 nutrients-15-02439-t003:** Odds ratio (95% confidence intervals) for obesity according to the dietary phytochemical index quartile stratified by sex.

	Boys (n = 623)					Girls (n = 573)				
	Q1 (n = 155)	Q2 (n = 156)	Q3 (n = 156)	Q4 (n = 156)	*p* Value/ *p* for Trend	Q1 (n = 143)	Q2 (n = 143)	Q3 (n = 144)	Q4 (n = 143)	*p* Value/ *p* for Trend
Weight (kg)	18.3 ± 90.68	17.98 ± 0.65	17.66 ± 0.69	17.51 ± 0.26	0.168 *	17.13 ± 0.65	17.48 ± 0.68	17.36 ± 0.64	17.07 ± 0.26	0.694
Energy (kcal/day)	1389 ± 97.6	1479 ± 96.9	1533 ± 97.5	1412 ± 40.4	0.061	1296 ± 95.4	1320 ± 90.7	1236 ± 86.0	1252 ± 35.9	0.380
Weight percentile ^#^										
Crude	1(ref)	0.790 ^†^ (0.311–2.009)	0.461 (0.153–13.395)	0.286 (0.094–0.866)	0.017 ^†^	1(ref)	2.128 (0.788–5.745)	1.033 (0.327–3.259)	0.956 (0.313–2.918)	0.468
Model 1	1(ref)	0.743 (0.286–1.928)	0.410 (0.141–1.192)	0.282 (0.094–0.850)	0.014	1(ref)	2.125 (0.782–5.774)	1.084 (0.337–3.493)	0.991 (0.320–3.072)	0.540
Model 2	1(ref)	0.826 (0.324–2.106)	0.467 (0.154–1.414)	0.293 (0.096–0.890)	0.018	1(ref)	2.128 (0.789–5.738)	1.032 (0.327–3.259)	0.955 (0.313–2.918)	0.468
Model 3	1(ref)	0.769 (0.295–2.009)	0.414 (0.142–1.212)	0.287 (0.095–0.868)	0.016	1(ref)	2.119 (0.783–5.730)	1.083 (0.336–3.489)	0.989 (0.319–3.064)	0.539
BMI percentile										
Crude	1(ref)	0.906 (0.371–2.21)	0.473 (0.157–1.423)	0.459 (0.148–1.143)	0.114	1(ref)	2.870 (0.964–8.541)	1.877 (0.578–6.099)	1.271 (0.360–4.486)	0.807
Model 1	1(ref)	0.850 (0.342–20,115)	0.419 (0.142–1.221)	0.454 (0.148–1.392)	0.102	1(ref)	2.883 (0.961–8.650)	2.016 (0.618–6.580)	1.341 (0.374–4.813)	0.917
Model 2	1(ref)	0.928 (0.382–2.255)	0.476 (0.158–1.438)	0.466 (0.151–1.437)	0.115	1(ref)	2.899 (0.966–8.698)	1.898 (0.581–6.201)	1.271 (0.358–4.510)	0.806
Model 3	1(ref)	0.861 (0.348–2.129)	0.419 (0.142–1.232)	0.458 (0.151–1.385)	0.101	1(ref)	2.962 (0.980–8.955)	2.047 (0.621–6.746)	1.354 (0.373–4.912)	0.920

DPI, dietary phytochemical index; Q, quartile; BMI, body mass index; BMI percentile, body mass index percentile. Model 1, adjusted for energy intake; Model 2, adjusted for age; Model 3, adjusted for energy intake and age; all the values are presented as mean ± standard deviation or odds ratio (95% confidence intervals). * Values are presented as *p*-values calculated using PROC SURVEYREG. ^#^ Values are presented as *p* for trends, calculated by logistic regression analysis using the PROC SURVEY procedure. ^†^ The weight and BMI percentiles were defined according to the Korean Society for the Study of Obesity criteria; we analyzed the odds ratio for obesity as having ≥95 percentiles.

## Data Availability

Not applicable.

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
