# Peer review of "Association between the Dietary Phytochemical Index and Lower Prevalence of Obesity in Korean Preschoolers"

_nutrients, 2023, doi:10.3390/nu15112439_

Round 1
Reviewer 1 Report
To investage correlation between dietary phytochemical index and lower prevalence of obesity in Korean preschoolers,the research is interesting. There is some confuse problems need to clear:
(1) Is BMI value suitable for children?
(2) We can not find preschoolers average height according different age. The value is important in this study.
(3) Authors mentioned some phytochemical food such as whole grains and etc. and list eggs, fishes as phytochemical food groups. I can understand authors purpose, but will misslead consumers. I hope authors can consider the perspective of nutritional balance. So I think the conclusion is too one-sided.
Author Response
- Is BMI value suitable for children?
- A child’s BMI must be interpreted relative to other children of the same sex and age, not with a single BMI score. This is because weight and height change during children’s growth and development, as does their relation to body fatness. The BMI-for-age percentile growth charts are the most commonly used indicator to measure the size and growth patterns of children and teens in Korea, as well as the United States. These percentiles are calculated from the Korea Centers for Disease Control and Prevention (KDCA) growth charts, which were collected since 1967. This information was presented in the manuscript (page 3, lines 102-109).
- We analyzed the prevalence of obesity according to the dietary phytochemical index quartile based on the BMI percentile provided by the Korea National Health and Nutrition Examination Survey (KNHANES). However, we also provided the BMI as well as the BMI percentiles, assuming it can help us understand the paper.
- I agree with the reviewer's comments that a single BMI value is not suitable for children. Therefore, we revised the manuscript by eliminating BMI values in Tables 1 and 3.
- We cannot find preschoolers average height according different age. The value is important in this study.
- Preschoolers’ average height and weight according to different age have been added to the manuscript (page 4, table 1).
- Authors mentioned some phytochemical food such as whole grains and etc. and list eggs, fishes as phytochemical food groups. I can understand authors purpose but will mislead consumers. I hope authors can consider the perspective of nutritional balance. So, I think the conclusion is too one-sided.
- In <Table 2>, we show eggs, fish, etc. as a list of non-phytochemical food groups. In addition, we have carefully checked whether there is anything that could cause misunderstanding among readers and modified some of the Discussion and Conclusion.

Reviewer 2 Report
The manuscript entitled “Association Between the Dietary Phytochemical Index and Lower Prevalence of Obesity in Korean Preschoolers” presents interesting issue, however some corrections are needed.
– Lines 108-19- “The 24-hour recall survey was conducted through face-to-face interviews with the trained staff. “ It were children aged 3–5 years – who completed the 24-hour recall survey for the children? Parents? What about food in kindergarten? Grandparents? Etc. Please specify it
– Taking into account the fact, that the study involvement minors, not only informed consent of minors but also from parents or legal guardians are requited. Please clarify it in the light of local regulations
– Lines 144-145 -‘Models 1, 2, and 3 were adjusted for energy intake, age, and energy intake and age, respectively’ - It's hard to tell what the models were different. Please be more precise.
– Table 1 - BMI (kg/m2); Height (cm); Weight (kg) – there is no need to present this data
– Table 1 – please defied ‘Household income” level
– Table 2 – is difficult to follow due to the some merging of numbers. I don't know if I see it right but I see huge difference in total DPI in boys and girls quartiles (e.g. 28.33 vs 64 for 4Q). Where does that difference come from?
– The discussion should be more closely related to the results. Authors should in their discussion include 3 areas: (1) compare gathered data with the results by other authors, (2) formulate implications of the results of their study and studies by other authors, (3) formulate the future areas which should be studied.
– Authors should discuss the limitations of their study!
Author Response
Please see the attachment.
- Lines 108-19- “The 24-hour recall survey was conducted through face-to-face interviews with the trained staff. “ It were children aged 3–5 years – who completed the 24-hour recall survey for the children? Parents? What about food in kindergarten? Grandparents? Etc. Please specify it
- Trained dietitians collected recalled dietary intake data in person and also gathered recipes for each food item consumed by respondents in the previous 24-hour time period. For children younger than 8 years of age, one of the parents answered the dietary recall. This information has been added to the manuscript with the reference (page 3, lines 117-118).
- Since the Korea National Health and Nutrition Examination Survey (KNHANES) we analyzed is a sample survey conducted through household visits, the subjects have time to prepare their answers in advance. In the case of children’s meals consumed outside the home, such as in kindergarten, parents can describe them accurately since the menu list may be provided by the institution.
- Taking into account the fact, that the study involvement minors, not only informed consent of minors but also from parents or legal guardians are requited. Please clarify it in the light of local regulations
- We analyzed the KNHANES which is a large-scale cross-sectional survey approved by the Institutional Review Board of the Korea Centers for Disease Control and Prevention (Approval numbers: 2013-07CON-03-4C, 2013-12EXP-03-5C, and 2018-01-03-P-A). Written informed consent was obtained from all the study participants to use and analyze their data. Detailed explanations about the survey have been added to the manuscript (page 2, lines 71-74).
- In order to analyze the KNHANES data and publish them in academic journals, we received a research ethics review at the research planning stage. This study was approved by the Institutional Review Board of the Seoul National University (E2108/002-002). This information was explained in the manuscript (page 2, lines 82-84).
- Lines 144-145 -‘Models 1, 2, and 3 were adjusted for energy intake, age, and energy intake and age, respectively’ - It's hard to tell what the models were different. Please be more precise.
- Detailed explanations of each models have been added to the manuscript to be more precise (page 4, lines 153-155).
- Table 1 - BMI (kg/m2); Height (cm); Weight (kg) – there is no need to present this data
- I agree with the reviewer's comments that a single BMI value is not suitable for children. A child’s BMI must be interpreted relative to other children of the same sex and age, not with a single BMI score. This is because weight and height change during children’s growth and development, as does their relation to body fatness. The BMI-for-age percentile growth charts are the most commonly used indicator to measure the size and growth patterns of children and teens in Korea, as well as the United States. This information was presented in the manuscript (page3, lines 102-109). Therefore, we revised the manuscript by eliminating BMI values in Tables 1 and 3.
- Furthermore, preschoolers’ average height and weight according to different age have been added to the manuscript (page 4, table 1) since these values may be important in this study.
- Table 1 – please defied ‘Household income” level
- Household income level was classified in quartiles (low, middle-low, middle-high, and high) provided by the KNHANES. Standard of household income level quartiles (unit: million won) was varied according to years of survey conducted as follows:
Survey year |
First q-quantile |
Second q-quantile |
Third q-quantile |
2013 |
75.00 |
150.00 |
246.31 |
2014 |
68.06 |
148.04 |
250.00 |
2015 |
76.12 |
157.59 |
268.77 |
2016 |
75.00 |
150.00 |
246.31 |
2017 |
89.44 |
190.57 |
310.42 |
2018 |
106.07 |
202.07 |
317.96 |
- Detailed explanations have been added to the manuscript (page 2-3, lines 94-96)
- Table 2 – is difficult to follow due to the some merging of numbers. I don't know if I see it right but I see huge difference in total DPI in boys and girls quartiles (e.g. 28.33 vs 64 for 4Q). Where does that difference come from?
- Total DPI by quartiles in boys is 5.12, 11.30, 17.09, 28.33, and 5.7, 11.88, 17.93, 29.64 for girls. I agree with the author’s idea that reading numbers in Table 2 is difficult because of little margins. However, I tried to follow the format instruction of Nutrients and hope there will be a chance to edit on the process of publishing.
- The discussion should be more closely related to the results. Authors should in their discussion include 3 areas: (1) compare gathered data with the results by other authors, (2) formulate implications of the results of their study and studies by other authors, (3) formulate the future areas which should be studied.
- I added the contents to the Discussion section and marked it in response to your feedback.
- Authors should discuss the limitations of their study!
- The limitations of the study were discussed in three ways in lines 302–313.

Round 2
Reviewer 1 Report
all comments has been revised.
Reviewer 2 Report
I appreciate the great efforts that the authors have made in response to my questions and concerns. I still have problem with Table 2 and 3 - is difficult to follow due to the some merging of numbers. The table is not readable, please provide more accurate version. Moreover, there are some small issue with formatting (see. Paragraph conclusion/ Author Contributions)